# The Effect of Color Saturation of Travel Pictures on Consumer Appeal

**Li Lin, Yuting Chen, Hong Zhu and Jiwang You ***

Business School, Nanjing University, Nanjing 210093, China; mf1932111@smail.nju.edu.cn (L.L.)
* Correspondence: mp20020418@smail.nju.edu.cn

**Abstract:** In the Internet era, online channels have contributed significantly to tourism marketing and promotion. Consumers will receive tourism information online to reduce information asymmetry. Moreover, with overwhelming levels of information, consumers may only get a limited amount of information. Therefore, the primary concern for marketers lies in capturing consumers' attention during this stage. Visual design is essential among the various factors researchers have thoroughly investigated. Studies have recognized the importance of the color characteristics of travel photos on consumer decisions. However, these studies often focus their analysis on tonal information while neglecting the intuitive emotional impact of color saturation on consumers. Consequently, this study aims to distinguish and categorize different tourist destinations to explore the impact of saturation characteristics of attraction photos on consumer appeal in online scenarios. This study contributes to existing knowledge of color context theory, enhancing its contextual application. The findings presented in this paper have implications for developing more effective visual marketing strategies.

**Keywords:** saturation; geographic distance; attractiveness; color psychology

## 1. Introduction

In the internet era, online channels have emerged as important communication channels for the tourism industry. According to the Meituan 2022 National Day Travel Report (https://mp.weixin.qq.com/s/q5u_Fo1ZXBfNAHoi4qX7Qg (accessed on 1 April 2023)), which highlights key findings and statistics related to travel behaviors, preferences, and destinations among Chinese tourists and provides insights and data on travel trends during China's National Day holiday in 2022, there was a significant increase in the search volume for keywords such as "local travel and peripheral travel," with a year-over-year surge of 440% observed in the week before the holiday. Consequently, obtaining e-WOM (electronic Word of Mouth) from the internet has become essential for consumers before traveling.

The critical role of online channels in tourism marketing campaigns stems from the information asymmetry between consumers and tourism destinations. Through online access to tourism information, consumers can effectively reduce information asymmetry and influence tourism consumption decisions [1,2]. Nevertheless, the affordability and accessibility of internet communication expose consumers to a substantial volume of pertinent information when seeking guidance, such as travel recommendations.

The information consumers can access is limited throughout a multitude of information streams. Therefore, the primary concern for marketers lies in determining effective strategies to capture the attention of consumers during this stage; thus, many researchers have focused on visual design to attract the attention of consumers online [3–5]. When conducting a quick information search, consumers have limited cognitive processing resources for each piece of information. At this point, they will tend to make decisions through intuitive perceptions rather than logical thinking, which requires much effort [6,7]. The information conveyed by visual design fulfills this consumer need. Consequently, content disseminated online is often accompanied by photos to attract consumers [8–10].

Color is a visual stimulus that contributes significantly to the decision-making of consumers, particularly when they are required to make immediate decisions [11]. Furthermore, color information is involved in tourism marketing; therefore, research on travel photo color represents a prominent area of interest. Although studies have noted the importance of travel photo color characteristics on consumer decision-making, these studies have often focused their analysis on hue information [12,13]. However, saturation significantly impacts consumers' intuitive, emotional perceptions [14]. Insufficient discussion regarding the other two dimensions of color, namely saturation, and luminance, for context-specific effects has resulted in different research findings. Color context theory states that the psychological meaning of color varies significantly across contexts [15]. For example, red has a warning meaning in competitive contexts, while it represents sexual attraction in relationship contexts [16,17]. The study concluded that a similar contextual effect exists in travel photo saturation.

Moreover, low saturation is matched with a more distant psychological distance, whereas high saturation is matched with a closer psychological distance [11,18,19]. This psychological distance may show different effects in different contexts and different types of tourist destinations. Herein, we have selected two common tourist destinations, cultural and natural, to discuss [12,13]. When tourist photo content is a nature-based attraction, high-saturation colors are correlated to closer psychological distance and can lead to more positive consumer responses [20]. Contrarily, when tourist photo content is a culture-based attraction, low saturation colors represent greater distance. This, in turn, enhances the attraction of culture-based attractions by highlighting their distance, thereby relatively diminishing the positive effect of high saturation.

This study aims to distinguish and categorize different types of tourist destinations to explore the effect of the saturation characteristics in attraction photos on consumer attractiveness in online scenarios. Specifically, we aim to compare cultural and natural tourist attractions to validate this effect and find the corresponding explanatory mechanisms. Our results can enrich the exploration of the impact of visual stimuli in tourism marketing. Particularly, the results will contribute to a deeper understanding of the significance of color saturation and propose the existence of an interaction effect between attraction features and color. This study also further enriches the contextual application of color context theory and can confirm that contextual factors influence the psychological meaning of color saturation and hue. Moreover, our results can provide a reference for tourist attractions to choose appropriate visual marketing strategies according to their characteristics.

## 2. Literature Review

### 2.1. Influence of Color on Consumer Evaluation

#### 2.1.1. Psychological Effect of Color

Color-related research in sensory marketing and psychology has predominantly focused on the HSV (Hue, Saturation, Value) color space model [21]. This model uses hue, saturation, and luminance to describe colors. The term "hue" in this context refers to the difference in the color name that is closely related to the color light wavelength. The observed change in hue, as the wavelength of light decreases within the visible spectrum, includes a sequence of colors, including red, orange, yellow, green, cyan, blue, and violet. Brightness represents the color brightness degree and refers to the proportion of colored light mixed with white or black light. The higher the brightness, the stronger the light, the brighter the color. Saturation is the color purity and intensity (also known as vividness) representing the proportion of gray light mixed with color light [22], researchers stated that the image's saturation was the most critical attribute perceived by the respondents, followed by caption description style, hue and brightness [23]. A higher degree of color purity is observed when the proportion of gray light combined with colored light is reduced, leading to increased vividness of the color. The HSV model is proposed based on the physical properties of colored light, yet its three dimensions align closely with the human intuitive perception of color, in contrast to color space models, including RGB

(Red, Green and Blue) and CMYK (Cyan, Magenta, Yellow and Black). Therefore, most studies exploring the psychological effects of color have adopted the HSV model. The blue-decorated stores were more likely to sell their goods than the red-decorated stores. This can be attributed to the relaxing and uplifting effect that blue has on individuals, as opposed to the anxiety-inducing impressions of red [24,25]. The effect of color on consumer evaluations is also supported by electrophysiological evidence. Specifically, red-packaged goods elicit heightened sympathetic nerve activity among consumers, associated with increased nervousness. Conversely, blue-packaged goods elicit reduced sympathetic nerve activity, associated with more relaxation. Consequently, consumers perceive red-packaged products as more likely to threaten their health [26]. Recent studies further enriched the downstream effects of physiological arousal on consumer product evaluations due to color. For instance, color differences affect consumers' price acceptance of products [4], durability evaluation [27], and immersion experience [28], among others.

Besides physiological arousal, color changes the cognitive patterns of consumers. Among them, the correlation between saturation and psychological distance is a primary research direction. Psychological distance refers to the distance between oneself and things perceived by a person-centered on oneself [29–31]. Psychological distance contains four main dimensions: temporal distance, spatial distance, social distance, and probability size [29]. A negative association exists between color saturation and psychological distance. The higher the saturation, the closer the psychological distance. This effect is supported by research on the spatial, temporal, and social dimensions of psychological distance.

Regarding spatial distance, in the early 1900s, it was suggested that reducing bright colors in artwork would make viewers feel further away from the work [32]. Moreover, people perceive highly saturated color objects as being closer to them or appearing larger [21,33]. Regarding temporal distance, Lee et al. observed that in contexts characterized by more temporal distance (e.g., recalling a hotel they stayed in years ago), consumers would perceive low-saturation photos as more consistent with their memories [18]. Regarding social distance, Xiao et al. found that high saturation logo colors could bring the brand closer to consumers. Consequently, this improves consumers' evaluation of the brand's warmth trait [19]. Briefly, the higher the saturation, the closer the psychological distance.

### 2.1.2. Contextuality of the Psychological Effect of Color

The color context theory is widely accepted among researchers for the influence of color on individuals' emotions and perceptions and memorization [34–36]. Contextual color theory suggests that the psychological effects of color stem from the connection between color and a particular meaning during innate evolutionary and sociocultural processes [15]. The context significantly affects the specific meaning of this connection. For instance, in human society, and depending on the specific context, red represents two different meanings: "warning" and "sexy" [16,17]. In competitive contexts, people often use red to indicate aggression or to express strong emotions such as anger [37]. Whereas, in relationship contexts related to mate choice, red represents sexiness or sexual openness [2,38]. Therefore, people may respond differently to the same color in different contexts. For instance, cross-cultural studies of general color preferences confirm that blue is a universally preferred color for humans, while brown and orange are universally disliked colors [39]. However, consumers can strongly reject blue meat foods. Moreover, studies on travel photos have found that the presence of brown and orange elements can significantly increase the likability for those taking photos of urban landscapes [13,40].

Briefly, attitudes of people toward color are determined mainly by specific contexts. Currently, when exploring the overall state of research, there is a predominant focus on investigating contextual effects of color with a particular emphasis on the hue dimension, particularly the comparison between red and other colors [41]. However, exploring saturation in this context has received comparatively less attention. This study can complement the research on color contextual effects by exploring the different effects of color saturation on tourist photos in different tourist destinations.

*2.2. Travel Photo Study*

The influence of travel photo color characteristics on consumer decision-making is one of the recent research hotspots. Studies demonstrated that a photo whose projected images matched consumers' central impressions, particularly affective ones, could induce higher online engagement [42]. By analyzing travel photos in the Zhuhai-Hong Kong-Macao Greater Bay Area, Yu et al. derived an association between color characteristics and the number of likes received [13]. The results revealed that photos containing orange, yellow, blue, and violet colors received more likes and comments. Regarding travel photos, blue and violet colors increased the likes for art or culture photos, while orange and yellow brought positive results in modern cityscape photos. An additional study, which similarly analyzed travel photos, highlighted that blue could increase the attractiveness of landscapes, ancient buildings, and food photos, while violet and some warm colors can increase the attractiveness of cityscape photos [12,43]. Finally, the findings of related studies on hue have predominantly exhibited a high degree of consistency. These studies also show that changes in photo content affect consumers' responses to different shades of color. However, the results of these studies varied regarding the remaining two aspects of color, namely saturation and luminance. Yu et al. found that luminance and saturation alone showed limited predictive ability in determining the number of likes received for travel photos. However, regarding food-related photographs, a positive correlation was observed only with saturation [13,44]. This result is consistent with previous studies in which people prefer highly saturated food [45]. Furthermore, Yu 2021 [12] found a strong correlation between hue and saturation; therefore, the role of the hue factor was mainly analyzed. In conclusion, there remains a shortage of research on the effect of color saturation in travel photos.

The color saturation of travel photos should be an essential factor influencing consumer attitudes. Saturation is more likely to affect consumers' intuitive perceptions of color stimuli han hue [19]. In the context of this study, i.e., the effect of online tourist attraction photos on consumer responses, consumers are confronted with a large amount of information, making them more inclined to make quick decisions through their intuition. This means that color saturation may contribute in this context. Color saturation also has a stronger practical value in promoting tourist attractions due to the fixed tonal characteristics of many attractions. For example, the walls of the National Palace Museum are mostly decorated with warm red walls and yellow tiles; blue and yellow tones often dominate the sea and beaches of Sanya. These tones, as core factors, are difficult to change. Therefore, when applied to the promotion of tourist attractions, the applicability of hue research may be more limited. Contrarily, saturation changes are relatively more flexible, allowing merchants or users to change photo saturation through various common photo editing apps. This finding enables them to choose between displaying a high-saturation vivid blue sea or a low-saturation blue sea. Consequently, this study chooses color saturation as an entry point to explore the differences in the attractiveness of attraction photos with different saturation levels to consumers in online scenarios.

## 3. Research Hypothesis

*3.1. The Role of Color Saturation on the Attractiveness of Travel Photos*

According to color context theory, the psychological effects of color may differ significantly in different contexts [15]. Therefore, the inconsistent results of prior research regarding the color saturation of tourism photos can be attributed to the failure of content segmentation in the context of color saturation studies in tourism photos. Particularly, the role of color saturation on tourism photo attractiveness may be reversed with changes in photo content. Psychological distance in the realm of tourism marketing exhibits a characteristic wherein its impact can be reversed with contextual changes. Therefore, this study focuses its theoretical analysis on the direct effect of color characteristics of tourist photos on consumers' evaluation of the attractiveness of the respective attraction. The study concluded that closer proximity, represented by high saturation, is more likely to

affect nature-based tourist attractions positively. Evolutionary psychology studies suggest that the human tendency to either approach or avoid some natural stimuli can be attributed to gradual adaptation to the natural environment throughout evolution [46]. For example, the tendency to approach clear water, clear skies, fresh food, and good views while avoiding rotten food or excrement comes from innate evolution rather than acquired learning [47]. Nature-based destinations attract tourists by displaying these elements that inspire convergence motivation [48]. Research has also confirmed that making visitors feel closer to nature and less distant from natural scenery can improve visitor ratings [49]. When saturation is associated with natural things, high saturation is often associated with those concepts that elicit convergence motivation.

Consequently, these associations represent positive psychological meaning [21,45,50] and reduce the sense of distance [21,33]. In food research, highly saturated colors often represent ripeness or freshness and can evoke consumer motivation to converge, thus improving food evaluation [21,45,50]. To summarize, this study infers that closer proximity, represented by high saturation, is consistent with tourists' convergence motivation for nature-based destinations and should result in positive effects.

However, the correlation between psychological distance and destination attractiveness is more complex for cultural destinations. The cultural dimension of psychological distance was found to have both adverse and positive effects on choosing the destination for the consumer. Among them, the positive factor is often associated with destination attractiveness; the distance brought by culture can bring a sense of novelty to tourists and positively attract them [51,52]. Moreover, cultural tourism programs are more attractive to tourists from other cultures [53]. However, distance can also have some adverse effects, which are focused on the actual tourist experience, such as inconvenience due to language barriers [51,54]. Generally, for cultural tourist attractions, psychological distance is a double-edged sword limiting the impact of creating or bringing closer psychological distance on the overall evaluation of a tourist destination.

Finally, this study establishes a correlation between color saturation and psychological distance and has found that higher color saturation corresponds to increased consumer attractiveness when the tourist photo content is a nature-based attraction. At the same time, the effect of color saturation is weakened due to psychological distance when the tourist photo content is a cultural-based attraction. Accordingly, the following hypotheses are proposed:

**H1a.** *For nature-based tourist attractions, high saturation color photos can improve attractiveness ratings.*

**H1b.** *For cultural-based attractions, the effect between saturation of color photos and attractiveness ratings disappears.*

**H2a.** *This effect is mediated by the psychological distance. For nature-based destinations, psychological distance perception reduces attractiveness.*

**H2b.** *For cultural-based destinations, the relationship between psychological distance and attractiveness is insignificant.*

### 3.2. Moderating Effect of the Geographical Distance of the Destination

This study argues that the difference in the effect of color saturation on natural and cultural attractions can be attributed to the different roles of distance perception. Accordingly, the study verifies that distance perception is the central factor contributing to this difference by directly manipulating distance perception. As mentioned, the perception of distance to a tourist destination among consumers can be influenced by various factors. Geographical distance is one of these prevalent and direct factors [55,56]. Local/foreign is a commonly used variable to manipulate geographic distance. Consumers have different preferences for the color saturation of logos of local and foreign companies [19]. Based on the previously discussed two-channel theory of information, in the context of travel photos, consumers

tend to rely on edge information when processing unfamiliar destination-related information and facing asymmetrical information. This preference for edge information stems from the challenges consumers encounter in making logical inferences through the core path. Color information is one type of marginal information. However, when consumers perceive themselves to possess a higher level of familiarity with the object being evaluated, they tend to develop a subjective belief that they can make logical inferences. Consequently, they tend more to prioritize the search for central path information while relatively neglecting peripheral or edge information.

According to this theory, when the attraction is in the local area, the limited information asymmetry between tourists and the attraction, coupled with the strong subjective familiarity of consumers, renders the reliance on color information in photos unnecessary for decision-making purposes. Therefore, this study concludes that the categorization effect of tourist destinations disappears for local destinations. Verifying that the moderating effect of geographic distance can support the mechanistic effect of distance perception and improve study validity. Accordingly, the following hypotheses are proposed in this study:

**H3a.** *Geographic distance has a moderating effect. For non-local destinations, low saturation is more attractive for cultural attraction photos, and high saturation is more attractive for natural attraction photos.*

**H3b.** *For local destinations, the positive effect of high saturation on cultural and natural attraction photos disappears.*

## 4. Data Collection and Research Methods

*4.1. Study 1: The Influence of Tourist Photo Saturation on Attractiveness and the Moderating Effect of Destination Type—An Experimental Approach*

First, the study used an experimental approach to directly verify the relationship between tourist photo saturation and attractiveness and the moderating effect of destination type. The reason for using this approach is that, when studying one dimension of color, it is necessary to control for the other dimensions. Herein, the experimental method allows more effective color hue and brightness control as well as directly examining the mediating effect of distance perception. Study 1 contains two sub-experiments with different contents of tourism photos (mainly natural and cultural landscapes) to exclude the effects caused by the photo content and improve the validity of the findings by repeating the experimental results.

The study used a two-saturation (high vs. low) × two-destination types (cultural vs. natural) between-subjects design (Figure 1). In total, 220 participants who have taken a real vacation before being surveyed were recruited from online social communities of a University in Nanjing. Participants completed the tasks online in exchange for a small payment. We used Sojump, a professional survey platform (https://www.sojump.com (accessed on 1 June 2023)) used in prior research [57] to record their responses. Six participants failed to pass the color blindness test, leaving the final sample size as 214 participants (40.65% female, $M_{age}$ = 26.38 years).

After being informed about the purpose of the research and study procedures, participants saw a scenic presentation with travel destination photos. The tourist destination photos were manipulated to be a cultural versus natural type. In order to mitigate the potential confounding effect of content similarity between the natural and cultural photo groups, a photo of a cliff was used. The only difference was that in the cultural group's photo, multiple grottoes were added to the cliff using Photoshop, and subjects would randomly see either a low or high-saturation photo (see Appendix A).

Moreover, the scenic introductory phrases for the two groups focused on introducing the cultural scenery ("Shiji Mountain Grotto Historical Scenic Area, feel the charm of the thousand-year-old grottoes") and the natural scenery ("Shiji Mountain Geopark, feel the charm of the strange peaks and rocks"), respectively, and participants were randomly shown one of the instructional phrases. Subsequently, participants were asked to rate

the attractiveness of the attraction on the Destination Attractiveness Scale. Destination attractiveness, which is considered to be closely related to tourists' emotions, has become a crucial factor for tourism destination managers and tourism researchers [58,59]. The participants were told to report their attractiveness to five items (for example: "The attraction is of interest to you"; "The attraction is attractive to you compared to other similar attractions"; 1 = "strongly disagree" and 7 = "strongly agree"; $r = 0.65$) (see Appendix B).

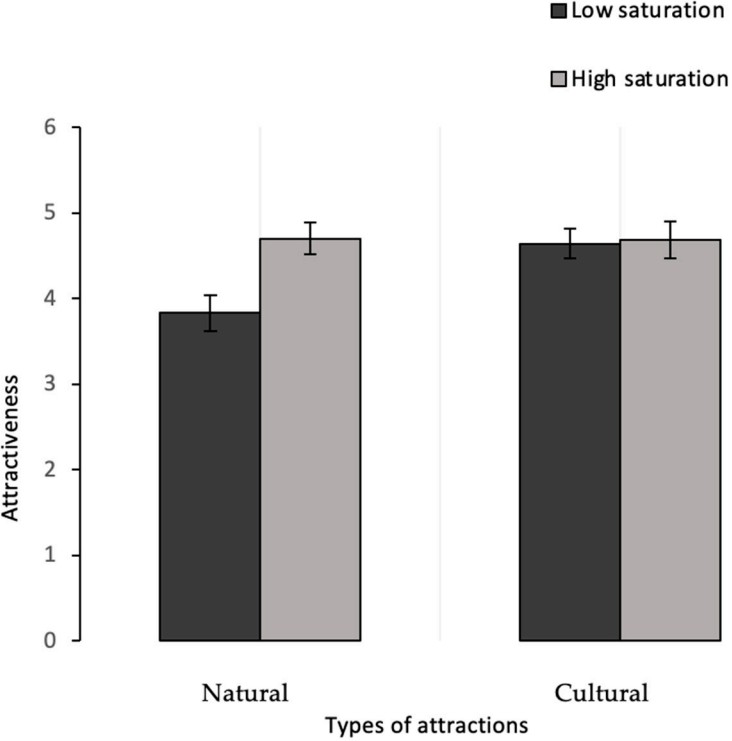

**Figure 1.** Interaction between type and saturation of tourism destinations on attractiveness. Source: Compiled by the author.

Destination Attractiveness Data Analysis

Herein, ANOVA (Analysis of Variance) was used to test the interaction effect of tourism destination type and tourism photo saturation in tourism destination attractiveness. The results found that the interaction effect of destination type and tourist photo saturation was significant ($F (1, 210) = 4.56$, $p = 0.034$). Additionally, the main effect of tourist photo saturation was significant ($F (1, 210) = 5.77$, $p = 0.017$, $M_{\text{high Saturation}} = 4.69 \pm 1.34$, $M_{\text{low saturation}} = 4.23 \pm 1.49$). Moreover, tourist destinations with high-saturation photos were more attractive. The main effect of destination type was significant ($F (1, 210) = 4.39$, $p = 0.037$, $M_{\text{cultural}} = 4.6\,7 \pm 1.39$, $M_{\text{natural}} = 4.27 \pm 1.46$).

Simple effects analysis showed that tourist photo saturation was significant for nature-based destinations. Specifically, it was observed that a higher saturation of brightly colored tourist photos effectively increased the destination attractiveness ($F (1, 210) = 10.49$, $p = 0.001$, $M_{\text{high saturation}} = 4.70 \pm 1.47$, $M_{\text{low saturation}} = 3.83 \pm 1.32$). In contrast, the effect of tourist photo saturation disappeared for cultural destinations ($F (1, 210) = 0.035$, $p = 0.852$, $M_{\text{high saturation}} = 4.69 \pm 1.22$, $M_{\text{low saturation}} = 4.64 \pm 1.54$).

### 4.2. Study 2: Mediating the Effect of Psychological Distance—An Experimental Approach

Study 1 validated the role of the effect of photo color saturation on various categories of tourist attractions. Study 2 expands upon the previous study by delving into the mechanistic role of psychological distance. In contrast, study 2 focused on tourism photos featuring buildings as their subject matter, as opposed to the mountain landscapes exam-

ined in study 1. Similar to the methodology employed in study 1, study 2 also employed textual descriptions to manipulate the types of attractions depicted in the photos.

The study used a two-saturation (high vs. low) × two-destination types (cultural vs. natural) between-subjects design (Figure 2). In total, 182 participants who did not participate in Study 1 and have taken a real vacation before being surveyed were recruited from online social communities of a University in Nanjing. Participants completed the tasks online in exchange for a small payment. We used Sojump, a professional survey platform (https://www.sojump.com (accessed on 23 April 2023)) used in prior research [57] to record their responses. Two participants failed to pass the color blindness test, leaving the final sample size as 180 participants (47.78% female, $M_{age}$ = 24.53 years).

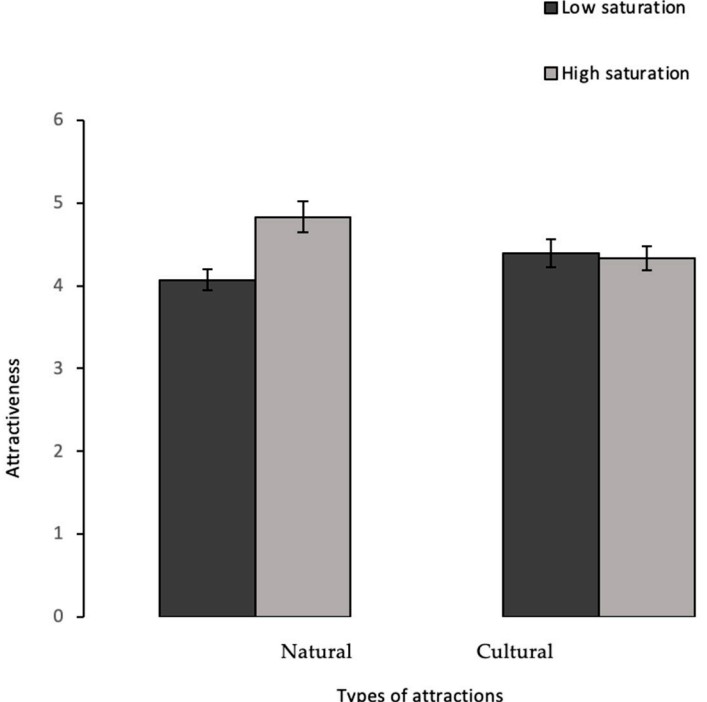

**Figure 2.** Interaction between type and saturation of tourism destinations featuring buildings on attractiveness. Source: Compiled by the author.

After being informed about the purpose of the research and study procedures, participants saw a scenic presentation with travel destination photos. Herein, the photo content was the same for the cultural and natural types, as both featured photos of an old village building in the mountains. This photo was processed into a high and low saturation version, and subjects were randomly shown either version of the photo (see Appendix A). Photos of tourist destinations were manipulated to be cultural versus natural types, as in study 1. The cultural type was described as "Dongxi ancient village, feel the charm of the ancient village", and the natural type was described as "Dongxi village, feel the charm of the green mountains and water". Subsequently, participants were asked to rate the attraction on both the Attractiveness of Tourist Destinations Scale [59] ("The attraction is attractive/good/wanted/interesting", 1 = "strongly disagree" and 7 = "strongly agree", $\alpha$ = 0.932) and the Perceived Distance Scale [60] ("I can feel as if the scenery in the photo is right in front of me"/"I feel close to the scenery in the photo", 1 = "strongly disagree" and 7 = "strongly agree", $r$ = 0.813).

### 4.2.1. Destination Attractiveness Data Analysis

Herein, ANOVA was used to test the interaction effect of tourism destination type and tourism photo saturation in tourism destination attractiveness. The results found a significant interaction effect of destination type and tourist photo saturation ($F$ (1, 176)

= 5.84, *p* = 0.017). The main effect of tourist photo saturation was significant (*F* (1, 176) = 4.11, *p* = 0.044, $M_{\text{high saturation}}$ = 4.58 ± 1.08, $M_{\text{low saturation}}$ = 4.23 ± 1.22). Furthermore, tourist destinations with high-saturation photos were more attractive. The main effect of destination type was non-significant (*F* (1, 176) = 0.27, *p* = 0.603, $M_{\text{cultural}}$ = 4.36 ± 1.12, $M_{\text{natural}}$ = 4.44 ± 1.21).

Simple effects analysis showed that for nature-based destinations, the effect of tourist photo saturation was significant, with high saturation of brightly colored tourist photos effectively increasing the destination attractiveness (*F* (1, 176) = 10.22, *p* = 0.002, $M_{\text{high saturation}}$ = 4.83 ± 0.90, $M_{\text{low saturation}}$ = 4.07 ± 1.35). Moreover, for cultural destinations, the effect of tourist photo saturation disappeared (*F* (1, 176) = 0.07, *p* = 0.786, $M_{\text{high Saturation}}$ = 4.33 ± 1.20, $M_{\text{low saturation}}$ = 4.39 ± 1.05).

### 4.2.2. The Mediating Role of Psychological Distance

The mediating role of psychological distance was analyzed using the Process 3.4 plug-in, selected Model 7, with a Bootstrap of 5000 times and a 95% confidence interval. The model used color saturation as the independent variable, destination attractiveness as the dependent variable, psychological distance as the mediating variable, and destination type as the moderating variable (Figure 3).

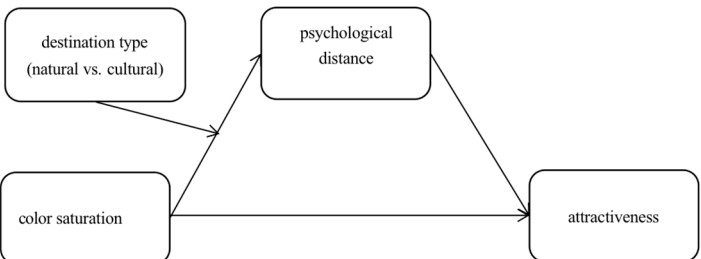

**Figure 3.** Research model. Source: Compiled by the author.

The results revealed that the overall model held 95% *CI* = (−0.82, −0.03), excluding zero. The interaction effects of color saturation and destination type on psychological distance were marginally significant [61] (*β* = −0.74, *t* (1, 176) = −1.95, *p* = 0.052). Moreover, the effect of psychological distance on destination attractiveness was significant (*β* = 0.47, *t* (1, 177) = 7.96, *p* < 0.001). Meanwhile, the effect of color saturation on destination attractiveness was marginally significant (*β* = −0.25, *t* (1, 177) = −1.71, *p* = 0.09). The overall direct effect was insignificant at 95% *CI* = (−0.14, −0.04), excluding zero. Additionally, the indirect effect of psychological distance was significant for natural category attractions (95% *CI* = (−0.63, −0.04), excluding zero). Simultaneously, the indirect effect of psychological distance was non-significant for human category attractions (95% *CI* = (−0.15, 0.33), including zero) (Figure 4). Briefly, the mediating effect of psychological distance holds only in the natural attractions category.

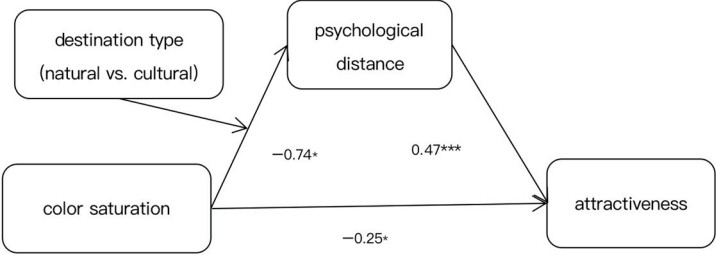

**Figure 4.** Research model with figure. Source: Compiled by the author. Note: * *p* < 0.1, *** *p* < 0.01.

*4.3. Study 3: The Influence of Tourist Photo Saturation on Consumer Views—A Secondary Data Study*

To improve the external validity of the findings, the study further used secondary data to verify whether there is a relationship between the color saturation of travel photos and consumer views in marketing practice. Previous secondary data studies in travel marketing have often used travel guide websites as data collection objects. Compared with social media platforms, consumers who browse travel guide websites generally have more apparent motivations for travel consumption and want to obtain relevant information. This is more appropriate in the intended research context to be explored in this study. Moreover, since travel tips are designed to address the information asymmetry between consumers and travel destinations, consumers who choose to consult travel tips are generally unfamiliar with these attractions. This is consistent with the finding in Study 2 that the effect exists only when the psychological distance is far. Therefore, the study first selected the travel guide website mafengwo for data collection. Among the platforms that treat travel tips as their core business, Ma Hive holds the largest market share (2019 data). Its market share is also only lower than that of Flying Pig and Go.com among all travel platform products (https://bg.qianzhan.com/trends/detail/506/210319-71517258.html (accessed on 1 May 2023)). Moreover, Ma Hive has a large volume, with about 30 million monthly active users in 2020 (https://qianfan.analysys.cn/sail/view/exquisite/index.html#/pageapp/pageapp?dE9wKzQ2amNTZ0lRT3JOV1RoQmxCdz09), and includes 600,000 travel tips (https://www.mafengwo.cn/).

Furthermore, these travel guides are accompanied by a prominent cover travel photo [62]. As mentioned earlier, the photo color is essential in quickly getting consumers' attention when searching for information. In summary, Hornet's Nest provides the ideal environment to verify the effect of color features on travel photos.

The study collected 5A scenic spots in the eastern provinces as the study population in January 2022. The reason for choosing the eastern provinces is that the geographical distribution of 5A scenic spots in the western region differs significantly between natural as well as cultural attractions. Since geographic features can significantly impact color characteristics, they are prone to systematic errors. The list of 5A scenic spots was obtained from the official website of the Ministry of Culture and Tourism in January 2022 (https://zwfw.mct.gov.cn/scenicspot (accessed on 1 July 202)). The study first excluded tourist destinations that could not be categorized as natural or cultural scenic spots, such as film and television bases, amusement parks, and modern buildings. It is worth noting that many of these attractions, including the Badaling Great Wall, possess cultural significance and natural beauty. Eventually, 33 attractions were identified for the final list. Therefore, the study invited consumers to rate the natural and cultural attractiveness of these attractions to select destinations with outstanding natural or cultural attractiveness. According to previous studies, similar rating classification tasks generally require about two people. Therefore, in this study, three consumers were recruited and told to rate the natural and cultural attractiveness of these attractions on a seven-point scale (1 = cultural category, 7 = natural category) [59,63]. The inter-rater reliability of the replicates ($\alpha = 0.814$) was relatively good. According to the general practice of previous work on classification tasks, attractions with mean values above the midpoint of the scale were selected as the natural attractiveness group. Whereas those below the midpoint of the scale were selected as the cultural attractiveness group. The Octopus tool, used by previous authors in secondary data of travel photos, was used to crawl 4143 related tips on Ma Hive with these attractions as keywords. The crawled content included cover photos of the tips, number of reads, author rating, travel days, and cost per capita. The study processed the cover photos through Python and used the same method as previous studies to calculate the color information. This was performed by extracting the saturation and luminance values of each pixel point in the photo and averaging them to obtain the color information of the photo. Since hue is a categorical variable in human perception, it is non-statistically significant to find the hue values of all pixel points (for example, the hue range of orange is 0~ 30°, the hue range

of purple is 240~270°, and the average value of both is 120~150°, which is the hue range of green; however, it does not make sense to consider green as the mean of orange and purple) [13]. In turn, abstracting representative colors would appear highly correlated between hue, saturation, and brightness.

Consequently, in later studies of travel photo colors, researchers kept only one of them [12]. Accordingly, this study followed this approach and put only saturation and luminance into the regression model. The number of reads represents how many consumers were attracted to the cheat sheet and thus clicked through to it. Therefore, the number of readers is used as the dependent variable. Author rank affects the attractiveness of the cheat sheet, so author rank is introduced as a control variable.

Data Analysis

The study used regression analysis to examine the association between tourist photo saturation and attractiveness. First, the study directly explored the association between saturation and clicks using simple linear regression analysis. The results in Table 1 shown that saturation could positively influence click-throughs overall ($\beta = 0.07$, $t = 4.49$, $p < 0.001$). This is consistent with previous studies in which bright colors attracted more attention. Subsequently, the study placed a regression model with tourist photo saturation and the moderating variable attraction type (0 = cultural category, 1 = natural category). The model results showed that there was still a significant positive relationship between tourist photo saturation and clicks ($\beta = 0.07$, $t = 4.52$, $p < 0.001$). Furthermore, attraction type directly influenced clicks ($\beta = -0.10$, $t = -6.51$, $p < 0.001$), increasing clicks for tips in the nature category. Subsequently, the study added cross terms of the independent and moderating variables into the regression model. The results showed that the regression coefficient for tourist photo saturation became negative at this point, but the effect was non-significant ($\beta = -0.07$, $t = -1.01$, $p = 0.055$); the effect of attraction type remained ($\beta = -0.17$, $t = -4.80$, $p < 0.001$). Specifically, the study found a significant interaction term between saturation and attraction type ($\beta = 0.16$, $t = 2.13$, $p = 0.033$), suggesting a moderating effect of attraction type on the effect of saturation and tourist photo attractiveness. After including author rank, brightness of travel photos, and pixel size of travel images as control variables, the interaction term remains significant ($\beta = 0.19$, $t = 2.71$, $p = 0.007$). Further analysis revealed that the relationship between saturation and clicks was insignificant for cultural attractions ($\beta = 0.01$, $t = 0.28$, $p = 0.780$). Whereas for natural attractions, the relationship between saturation and clicks was positive, with higher photo saturation being more attractive to consumers ($\beta = 0.10$, $t = 5.54$, $p < 0.001$). Overall, these results support the research hypothesis.

**Table 1.** Relationship between tourist photo saturation and attractiveness.

| | Dependent Variable: Attractiveness (Number of Clicks) | | | | | | | |
|---|---|---|---|---|---|---|---|---|
| | Model 1 | | Model 2 | | Model 3 | | Model 4 | |
| | B | SE | B | SE | B | SE | B | SE |
| **Variables** | | | | | | | | |
| Saturation | 0.070 *** | 0.02 | 0.070 *** | 0.02 | −0.07 * | 0.07 | −0.11 * | 0.06 |
| Type of attraction (0= cultural, 1 = natural) | | | −0.10 *** | 0.02 | −0.17 *** | 0.04 | −0.13 *** | 0.03 |
| Saturation × Type of attraction | | | | | 0.16 ** | 0.08 | 0.19 *** | 0.07 |
| **Control variables** | | | | | | | | |
| Author Level | | | | | | | 0.239 *** | 0.02 |
| Travel Photo Brightness | | | | | | | −0.012 | 0.01 |
| Travel photo pixel size | | | | | | | −0.052 *** | 0.01 |

Note: * $p < 0.1$, ** $p < 0.05$, *** $p < 0.01$. Source: Compiled by the author.

### 4.4. Study 4: The Interaction Effect of Geographical Distance on Different Types of Tourist Photo Saturation—An Experimental Approach

Studies 1 and 2 verified the moderating role of destination type (cultural vs. natural) on the relationship between tourist photo saturation and attractiveness. In particular,

study 2 also verified the mediating role of psychological distance. In marketing practices, geographic distance is an important and common factor in making consumers feel distant. Therefore, study 4 aimed to explore the interaction effect of the role of geographical distance and different types of tourist photo saturation. However, study 4 changed how the attractiveness of travel photos was administered to make the consumer's decision-making task closer to the real scenario, thus improving the validity of the study findings.

The study used a between-subjects design of two-destination types (cultural vs. natural) × two-geographical distances (far vs. near). In total, 199 participants who did not participate in Study 1 and 2 and have taken a real vacation before being surveyed were recruited from online social communities of a University in Nanjing. Participants completed the tasks online in exchange for a small payment. We used Sojump, a professional survey platform (https://www.sojump.com(accessed on 20 July 2023)) used in prior research [57] to record their responses. Three participants failed to pass the color blindness test, leaving the final sample size as 196 participants (50.0% female, $M_{age}$ = 23.73 ± 4.81 years).

After being informed about the purpose of the research and study procedures, participants saw a travel guide cover photo. The saturation manipulation was altered to allow consumers to select either a travel guide cover photo with low or high saturation. The rationale behind this step is that when consumers choose a low saturation cover photo, it signifies a higher attractiveness for the low saturation cover photo, and the converse. This choice has also been used in previous studies to verify the difference in the attractiveness of different visual stimuli to consumers [64]. The experimental procedure began by asking subjects to imagine preparing for a trip. Subsequently, half of the participants were assigned to read a natural attraction description, while the other half read a cultural attraction. These scenic presentations were randomly processed to either a high or low mental distance version. Then, they will see four travel guides; two random guides were treated as high-saturation and the other as low-saturation. The content of the photos was similar. Subjects were asked to select one of the guides to read. After they selected the guide, all subjects were shown the same guide content and rated the attractiveness of the attraction (the measure was the same as in Experiment 1). Finally, participants were required to report demographic information (see Appendix A).

Data Analysis

The study used binary logistic regression to analyze the effect of destination type and geographic distance on consumer color saturation preferences. The subjects were coded according to the color saturation of the cheat sheet cover photo they selected (0 = select low saturation, 1 = select high saturation). The study results found in Table 2 that both the type of travel destination and geographic distance, as well as the interaction term between the two, predicted the subjects' saturation choices. The interaction between tourism destination type and geographic distance was significant ($b$ = 1.58, $SE$ = 0.61; $\chi^2$ (1) = 6.70, $p$ = 0.010). In the out-of-town tourism context, subjects had significantly stronger preferences for high-saturation cover map travel tips in the nature-based attraction context (79.17%) relative to the culture-based attraction (50.0%) ($b$ = −1.33, $SE$ = 0.45; $\chi^2$ (1) = 8.64, $p$ = 0.003). In contrast, the difference in the attractiveness of the high-saturation cover map was non-significant when subjects were in the cultural type of attraction (56.00%) context versus the natural type of attraction (50.00%) context in the local tourism context ($b$ = 0.241, $SE$ = 0.41; $\chi^2$ (1) = 0.354, $p$ = 0.552). The main effect of tourism destination type was significant ($b$ = −1.34, $SE$ = 0.45; $\chi^2$ (1) = 8.63, $p$ = 0.003), consistent with previous findings that the attractiveness of high-saturation cover maps was significantly higher in natural category contexts (64.58%) relative to the human category (53.00%). The main effect of travel destination distance was significant ($b$ = −1.34, $SE$ = 0.46; $\chi^2$ (1) = 8.50, $p$ = 0.004), with consumers preferring high-saturation cover photos in out-of-town travel contexts (64.58%) relative to peripheral travel (53.06%).

**Table 2.** Relationship between type of tourism destination, geographical distance, and saturation.

| | Model 1 | | Model 2 | | | |
| | | | Field | | Local | |
| | **B** | **SE** | **B** | **SE** | **B** | **SE** |
| --- | --- | --- | --- | --- | --- | --- |
| **Variables** | | | | | | |
| Types of attraction | −1.34 *** | 0.45 | −1.33 *** | 0.45 | 0.24 | 0.41 |
| Type of attraction × Geographic distance | | | 1.58 ** | 0.61 | 1.58 ** | 0.61 |

Note: ** $p < 0.05$, *** $p < 0.01$. Source: Compiled by the authors.

## 5. Research Findings and Insights

### 5.1. Research Findings

Four studies have explored the association of tourist photo color saturation with attractiveness. Study 1 found that higher tourist photo color saturation had a positive effect on nature-based destinations, but an adverse effect on cultural destinations which is similar to the earlier studies that found tourists tended to appreciate natural farm images containing vivid colors such as farm landscape [13] Study 1 confirms H1a and H1b. Study 2 also found that perceived distance mediated this effect, confirming H2a, H2b, H3a and H3b. Study 3 adds to this by adding secondary data from real marketing contexts. The results found that for cultural attractions, the lower the photo saturation, the higher the number of clicks on the guide. For natural attractions, the higher the saturation, the higher the number of clicks on the guide. Support for H1a and H1b was provided from real consumer responses. Based on this, study 4 found that for local attractions with a closer sense of distance, consumers always have a closer psychological distance to the attractions when the positive effect of high saturation disappears for natural and cultural attractions. However, for non-local attractions, high saturation still has an increased effect on the attractiveness of nature-based destinations.

### 5.2. Theoretical Contributions

The study has the following three main theoretical contributions. First, the study complements the color effect on tourist photos by exploring the interaction effect between the tourist attraction type and the tourist photo saturation. In previous studies, insufficient discussion of contextual segmentation has led to different conclusions on color saturation. This study provides a potential explanation for the inconsistent results in previous studies by exploring the mechanisms and boundaries. The study found that saturation produced an opposite effect for natural versus cultural attractions. Moreover, this effect was only valid for non-local attractions.

In contrast, local attractions consistently exhibit a positive effect of high saturation for local attractions due to consumers' greater familiarity and sense of proximity. Therefore, inconsistent results will likely be produced if the types of attractions are not segmented in these respects. This study provides a basis for segmenting attractions that complement the study of tourist photo saturation.

Second, the study further extends the inquiry into the psychological effects of color saturation. Although color saturation has received some research attention recently, the research is generally far less complete than that on hue. Particularly, current saturation research has mostly been limited to exploring the psychological role of saturation in enhancing emotional arousal and has less often addressed the psychosocial significance of saturation in different contexts. This also leaves the most widely accepted theory in color psychology research, color context theory, lacking empirical support from the saturation dimension. However, we find that saturation has opposite effects in the two contexts by comparing natural and cultural tourist attractions, confirming that the color context theory still holds in the saturation dimension, providing support to generalize the theory.

Finally, the study also explores the role of distance perception in tourism management. In previous studies, researchers have begun to realize that consumers' psychological distance to tourist destinations has a multidimensional character. This study supports

interconversion and interaction between different dimensions of distance perception by comparing natural and cultural destinations. In particular, this study links the visual features of tourist photos with consumers' perceptions of distance to tourist destinations. Our results revealed that this distance from visual features could be interpreted by consumers as different dimensions of distance depending on the context, enriching the discussion on the influencing factors and downstream effects of traveler-destination distance perception.

*5.3. Practical Implications*

This study provides guidance and reference for tourism marketing and promotion practices. Color saturation is an important part of the visual photo of products and brands. In the current information-explosive social media era, quickly attracting consumers through visual messages has become particularly important [65]. Our findings highlighted the importance of employing differentiated color design strategies in promotional campaigns, particularly when utilizing tourism photos to attract consumers. Specifically, it is recommended to employ visual elements with high color saturation for natural tourist attractions while opting for visual elements with low saturation for cultural tourist attractions.

At present, the role of UGC in marketing and promoting tourist attractions should not be underestimated. Marketers can use the official social media accounts of attractions to retweet, like, invite head traffic, and other ways to attract traffic to UGC. In this process, marketers can choose the appropriate saturation photos to improve the attractiveness of UGC content.

Moreover, we found that the relationship between saturation and attractiveness also changes for local versus non-local attractions. This finding also provides implications for segmenting tourist destinations and tailoring communication programs for differentiation. When a destination primarily appeals to local tourists or a "neighborhood" destination, natural and cultural tourist attractions can consider using more saturated visual materials for their campaigns. However, when the destination targets non-local visitors or wants to expand the proportion of non-local visitors, the color saturation characteristics can be adjusted according to the destination type.

*5.4. Research Limitations and Prospects*

This study distinguishes the relationship between the color saturation of tourism photos and the attractiveness of nature-based and cultural-based destinations. The study combines experimental methods with secondary data and focuses on what travel photo types are more attractive to consumers at the early stage of consumer choice. Future research can further build on this foundation by exploring some behaviors after attracting consumers. These behaviors entail whether these people choose to travel to these places after looking at promotional photos with different saturation levels, whether they feel that the publicity is consistent with their feelings after traveling, and their willingness to post comments on travel websites. These conversion behaviors after being attracted can be the focus of future research. In addition, this study focused on the overall saturation of the color of travel photos. Future research could build on this and segment the content of the travel photos themselves. Various factors, including the perspective of the photo, the placement of people and objects within the photo, and the size of the product, may influence the evaluation of consumers. Further research can be conducted to investigate the potential interaction effect between these factors and color saturation to enhance the understanding of visual factors in tourism photos.

**Author Contributions:** Resources, J.Y.; Data curation, L.L.; Writing—original draft, L.L.; Writing—review & editing, Y.C. and H.Z.; Project administration, L.L.; Funding acquisition, J.Y. All authors have read and agreed to the published version of the manuscript.

**Funding:** This research received no external funding.

**Institutional Review Board Statement:** Ethical review and approval were not applicable in this research.

**Informed Consent Statement:** Informed consent was obtained from all subjects involved in the study.

**Data Availability Statement:** Ethical review and approval are not applicable.

**Conflicts of Interest:** The authors declare no conflict of interest.

## Appendix A. Materials Used in Studies

Stimuli for Study 1:

natural photo groups:

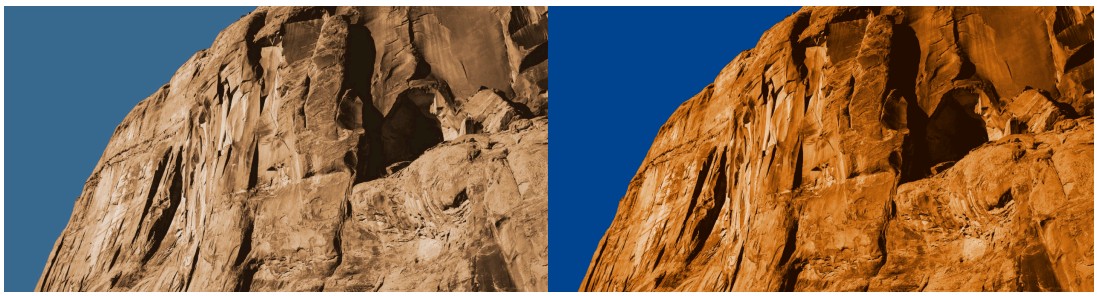

low-saturation                              high-saturation

cultural photo groups:

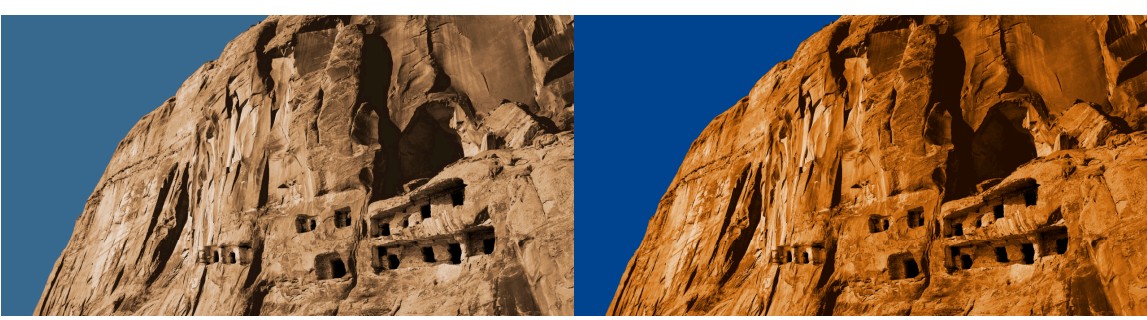

low-saturation                              high-saturation

**Figure A1.** Study materials to explore type and saturation of tourism destinations on attractiveness.

Stimuli for Study 2:

natural photo groups:

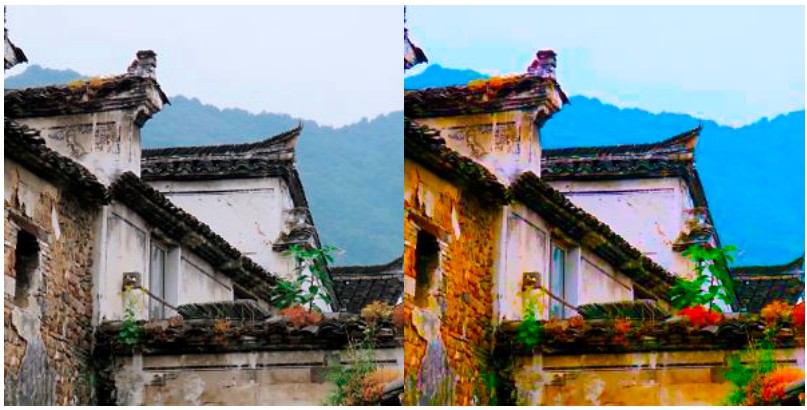

low-saturation                    high-saturation

**Figure A2.** *Cont*.

cultural photo groups:

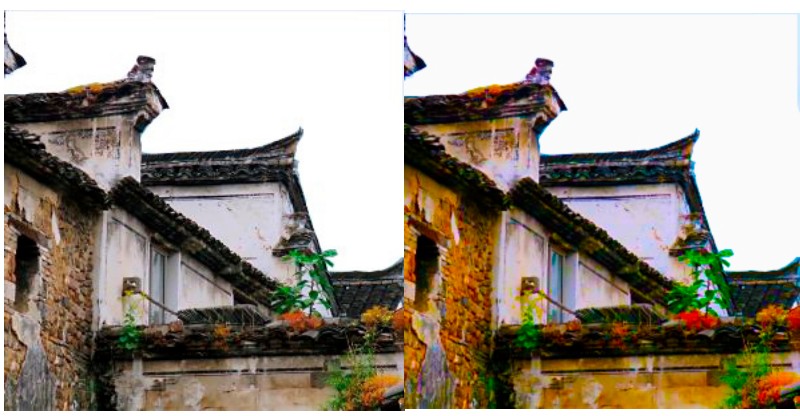

low-saturation                    high-saturation

**Figure A2.** Study materials to explore type and saturation of tourism destinations featuring buildings on attractiveness.

Stimuli for Study 4:

cultural photo groups:              natural photo groups:

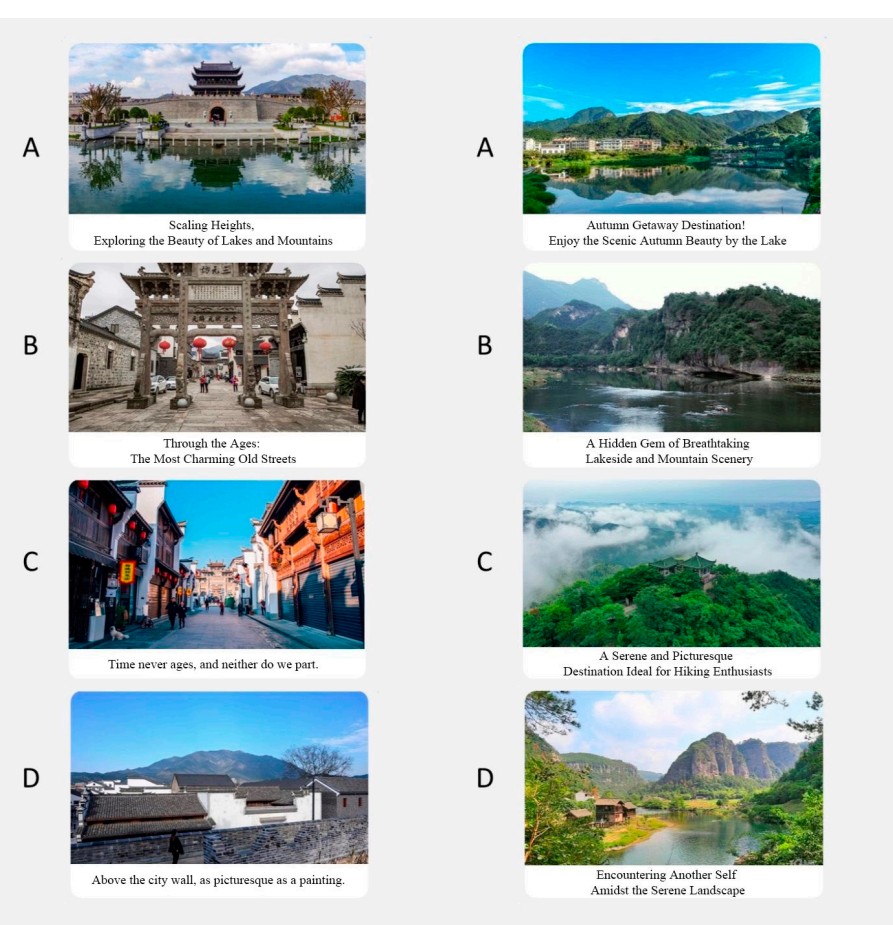

**Figure A3.** Study experiment to explore the interaction effect of the role of geographical distance and different types of tourist photos saturation (The left are four cultural photos and the right are four natural photos, the participants will only see four images on either the left or right side and then proceed to the next step).

**Appendix B**

Study 1

Thank you for participating in this survey.

Attention please:

1. Please read the following pictures and text carefully and answer according to your first impression and real thoughts.
2. All the questions and answers do not involve the examination of personal knowledge ability.
3. The questionnaire is completely anonymous and the answers you provide are for academic research only.

- Imagine that you have decided to travel. When looking for a travel guide, you noticed the attraction below.

  Please answer the following questions based on your intuition.

- The attraction is of interest to you.

  1 strongly disagree (1)
  2 (2)
  3 (3)
  4 (4)
  5 (5)
  6 (6)
  7 strongly agree (7)

- The attraction is attractive to you compared to other similar attractions.

  1 strongly disagree (1)
  2 (2)
  3 (3)
  4 (4)
  5 (5)
  6 (6)
  7 strongly agree (7)

- I am very interested in the scenic spot.

  1 strongly disagree (1)
  2 (2)
  3 (3)
  4 (4)
  5 (5)
  6 (6)
  7 strongly agree (7)

- The view of the scenic spot attracted me.

  1 strongly disagree (1)
  2 (2)
  3 (3)
  4 (4)
  5 (5)
  6 (6)
  7 strongly agree (7)

- The landscape in the photo attracted me.

  1 strongly disagree (1)
  2 (2)
  3 (3)

4 (4)
5 (5)
6 (6)
7 strongly agree (7)
Please answer the following questions based on the photo of scenic spot.
The Stone step Mountain is:

- Attractive.

  1 strongly disagree (1)
  2 (2)
  3 (3)
  4 (4)
  5 (5)
  6 (6)
  7 strongly agree (7)

- Good.

  1 strongly disagree (1)
  2 (2)
  3 (3)
  4 (4)
  5 (5)
  6 (6)
  7 strongly agree (7)

- Desirable.

  1 strongly disagree (1)
  2 (2)
  3 (3)
  4 (4)
  5 (5)
  6 (6)
  7 strongly agree (7)

- Diverting.

  1 strongly disagree (1)
  2 (2)
  3 (3)
  4 (4)
  5 (5)
  6 (6)
  7 strongly agree (7)

- Interesting.

  1 strongly disagree (1)
  2 (2)
  3 (3)
  4 (4)
  5 (5)
  6 (6)
  7 strongly agree (7)
Please answer the following questions based on your intuition.

- I could feel the scenery in the photo as if it were right in front of me.

  1 strongly disagree (1)
  2 (2)
  3 (3)
  4 (4)
  5 (5)
  6 (6)
  7 strongly agree (7)

- I felt myself very close to the scene in the picture.

  1 strongly disagree (1)
  2 (2)
  3 (3)
  4 (4)
  5 (5)
  6 (6)
  7 strongly agree (7)

- I can feel the traces of time here from the pictures.

  1 strongly disagree (1)
  2 (2)
  3 (3)
  4 (4)
  5 (5)
  6 (6)
  7 strongly agree (7)

- I can feel the time distance from the scenic spot.

  1 strongly disagree (1)
  2 (2)
  3 (3)
  4 (4)
  5 (5)
  6 (6)
  7 strongly agree (7)

- I felt distant from the landscape in the photo.

  1 strongly disagree (1)
  2 (2)
  3 (3)
  4 (4)
  5 (5)
  6 (6)
  7 strongly agree (7)

- I feel an emotional connection to the scenic spot.

  1 strongly disagree (1)
  2 (2)
  3 (3)
  4 (4)
  5 (5)
  6 (6)
  7 strongly agree (7)

- Gender

  Male (1)
  Female (2)

- Age

  ________________________________________________________

- What is the number in the picture

  ________________________________________________________

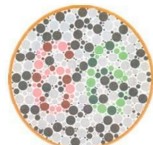

Study 2
Thank you for participating in this survey.
Attention please:

1. Please read the following pictures and text carefully and answer according to your first impression and real thoughts.
2. All the questions and answers do not involve the examination of personal knowledge ability.
3. The questionnaire is completely anonymous and the answers you provide are for academic research only.

- Imagine that you have decided to travel around for this weekend. When looking for a travel guide, you noticed the attraction below.

  Please answer the following questions based on your intuition.

- The scenic spot is very attractive to me.

  1 strongly disagree (1)
  2 (2)
  3 (3)
  4 (4)
  5 (5)
  6 (6)
  7 strongly agree (7)

- Compared with other similar attractions, this spot is very attractive to you.

  1 strongly disagree (1)
  2 (2)
  3 (3)
  4 (4)
  5 (5)
  6 (6)
  7 strongly agree (7)

- I am very interested in the scenic spot.

  1 strongly disagree (1)
  2 (2)
  3 (3)
  4 (4)
  5 (5)
  6 (6)
  7 strongly agree (7)

- The view of the scenic spot attracted me.

  1 strongly disagree (1)
  2 (2)
  3 (3)
  4 (4)
  5 (5)
  6 (6)
  7 strongly agree (7)

- The landscape in the photo attracted me.

  1 strongly disagree (1)
  2 (2)
  3 (3)
  4 (4)
  5 (5)
  6 (6)
  7 strongly agree (7)
  Please answer the following questions based on the photo of scenic spot.
  The Dongxi ancient Village is:

- Attractive.

  1 strongly disagree (1)
  2 (2)
  3 (3)
  4 (4)
  5 (5)
  6 (6)
  7 strongly agree (7)

- Good.

  1 strongly disagree (1)
  2 (2)
  3 (3)
  4 (4)
  5 (5)
  6 (6)
  7 strongly agree (7)

- Desirable.

  1 strongly disagree (1)
  2 (2)
  3 (3)
  4 (4)
  5 (5)
  6 (6)
  7 strongly agree (7)

- Diverting.

  1 strongly disagree (1)
  2 (2)
  3 (3)
  4 (4)
  5 (5)
  6 (6)
  7 strongly agree (7)

- Interesting.

1 strongly disagree (1)
2 (2)
3 (3)
4 (4)
5 (5)
6 (6)
7 strongly agree (7)
Please answer the following questions based on your intuition.

- I could feel the scenery in the photo as if it were right in front of me.

1 strongly disagree (1)
2 (2)
3 (3)
4 (4)
5 (5)
6 (6)
7 strongly agree (7)

- I felt myself very close to the scene in the picture.

1 strongly disagree (1)
2 (2)
3 (3)
4 (4)
5 (5)
6 (6)
7 strongly agree (7)

- I can feel the traces of time here from the pictures.

1 strongly disagree (1)
2 (2)
3 (3)
4 (4)
5 (5)
6 (6)
7 strongly agree (7)

- I can feel the time distance from the scenic spot.

1 strongly disagree (1)
2 (2)
3 (3)
4 (4)
5 (5)
6 (6)
7 strongly agree (7)

- I felt distant from the landscape in the photo.

1 strongly disagree (1)
2 (2)
3 (3)
4 (4)
5 (5)
6 (6)
7 strongly agree (7)

- I feel an emotional connection to the scenic spot.

  1 strongly disagree (1)
  2 (2)
  3 (3)
  4 (4)
  5 (5)
  6 (6)
  7 strongly agree (7)

- Gender

  Male (1)
  Female (2)

- Age

  _______________________________________________________________

  What is the number in the picture

  _______________________________________________________________

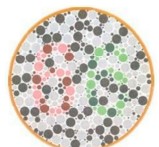

  Study 4
  Thank you for participating in this survey.
  Attention please:

1. Please read the following pictures and text carefully and answer according to your first impression and real thoughts.
2. All the questions and answers do not involve the examination of personal knowledge ability.
3. The questionnaire is completely anonymous and the answers you provide are for academic research only.

- Imagine searching for "recommendation of travelling around" on a social APP and seeing the following tweets.

  Please rate the cover image of the tweet you selected.

- The scenic spot is very attractive to me.

  1 strongly disagree (1)
  2 (2)
  3 (3)
  4 (4)
  5 (5)
  6 (6)
  7 strongly agree (7)

- Compared with other similar attractions, this spot is very attractive to you.

  1 strongly disagree (1)
  2 (2)
  3 (3)
  4 (4)
  5 (5)
  6 (6)
  7 strongly agree (7)

- I am very interested in the scenic spot.

  1 strongly disagree (1)
  2 (2)
  3 (3)
  4 (4)
  5 (5)
  6 (6)
  7 strongly agree (7)

- The view of the scenic spot attracted me.

  1 strongly disagree (1)
  2 (2)
  3 (3)
  4 (4)
  5 (5)
  6 (6)
  7 strongly agree (7)

- The landscape in the photo attracted me.

  1 strongly disagree (1)
  2 (2)
  3 (3)
  4 (4)
  5 (5)
  6 (6)
  7 strongly agree (7)
  Please answer the following questions based on your intuition.

- I could feel the scenery in the photo as if it were right in front of me.

  1 strongly disagree (1)
  2 (2)
  3 (3)
  4 (4)
  5 (5)
  6 (6)
  7 strongly agree (7)

- I felt myself very close to the scene in the picture.

  1 strongly disagree (1)
  2 (2)
  3 (3)
  4 (4)
  5 (5)
  6 (6)
  7 strongly agree (7)

- I can feel the traces of time here from the pictures.

  1 strongly disagree (1)
  2 (2)
  3 (3)
  4 (4)
  5 (5)
  6 (6)
  7 strongly agree (7)

- I can feel the time distance from the scenic spot.

  1 strongly disagree (1)
  2 (2)
  3 (3)
  4 (4)
  5 (5)
  6 (6)
  7 strongly agree (7)

- I felt distant from the landscape in the photo.

  1 strongly disagree (1)
  2 (2)
  3 (3)
  4 (4)
  5 (5)
  6 (6)
  7 strongly agree (7)

- I feel an emotional connection to the scenic spot.

  1 strongly disagree (1)
  2 (2)
  3 (3)
  4 (4)
  5 (5)
  6 (6)
  7 strongly agree (7)

- Gender

  Male (1)
  Female (2)

- Age

  _______________________________________________

  What is the number in the picture

  _______________________________________________

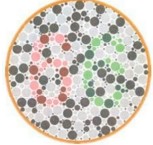

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
