# Peer review of "The Effect of Color Saturation of Travel Pictures on Consumer Appeal"

_sustainability, doi:10.3390/su151914503_

Round 1

Reviewer 1 Report

We recommend to the author/s:

- to revise the research hypothesis formulation - to present it more clearly. For example -  This effect is mediated by the sense of distance (but neither Geographic distance has a moderating effect) it is incomplete. Also..The specific path is that the higher the saturation, the lower the distance perception.

- present the Appendix, it is missing even mentioned in the text; therefore, the research data can not be checked and assessed;

- to specify the research sampling method;

- to describe the research sample, the demographic information and if this information was essential/relevant for the research;

- to explain the representativeness of the research sample;

- to specify how the online questionnaire was applied - was this questionnaire sent by email to potential respondents or posted on a specific platform? How was the access to the email addresses of the respondents?;

- to specify if the  Informed Consent was applied to participants and the content of it (it is not sufficient to say that the respondents were paid after completing the questionnaire online;

- to explain better the rating scale - Destination Attractiveness Scale and the value -  r = 0.65; without more explanations of these elements, the statistical tests are not supported;

- to explain the values -  α = 0.932,  r = 0.813....significance...appropriateness of the level (comparing with the theoretical level), etc;

- to explain the research findings in the light of other research/studies - are these findings in accordance or not with the results of other studies?;

- to make some corrections M 年龄 = 26.38?!

- to mention the source of each Figure and each Table;

- to explain the legend of Figure 2; even if it is mentioned in Figure 2- humanistic! There is not enough correlation with the explanations provided in the text;

- to follow the journal's guidelines in citing the references in the text! The text's citing style does not respect the journal's citing guidelines.

Author Response

Thank you very much for taking the time to review this manuscript. 

These are my responses to the comments:

Comment1:to revise the research hypothesis formulation - to present it more clearly. For example -  This effect is mediated by the sense of distance (but neither Geographic distance has a moderating effect) it is incomplete. Also..The specific path is that the higher the saturation, the lower the distance perception.

Response1:I have changed the hypothesis:

H1a: For nature-based tourist attractions, high saturation color photos can improve attractiveness ratings.

H1b: For cultural-based attractions, the effect between saturation color photos and attractiveness ratings disappears.

H2a: This effect is mediated by the psychological distance. For nature-based destinations, psychological distance perception reduces attractiveness.

H2b: For cultural -based destinations, the relationship between psychological distance and attractiveness is insignificant.

H3a: Geographic distance has a moderating effect. For non-local destinations, low saturation is more attractive for cultural attraction photos, and high saturation is more attractive for natural attraction photos.

H3b: For local destinations, the positive effect of high saturation on cultural and natural attraction photos disappears.

Comment2: present the Appendix, it is missing even mentioned in the text; therefore, the research data can not be checked and assessed;
Response2: I have added the appendix in the end of the thesis.

Comment3-7: 

to specify the research sampling method;

to describe the research sample, the demographic information and if this information was essential/relevant for the research;

to explain the representativeness of the research sample;

to specify how the online questionnaire was applied - was this questionnaire sent by email to potential respondents or posted on a specific platform? How was the access to the email addresses of the respondents?;

to specify if the  Informed Consent was applied to participants and the content of it (it is not sufficient to say that the respondents were paid after completing the questionnaire online;

Response 3-7: 

I have revised the research sampling in the 3.Data collection and research methods. 

Comment 8-9:

to explain better the rating scale - Destination Attractiveness Scale and the value r = 0.65; without more explanations of these elements, the statistical tests are not supported;

to explain the values -  α = 0.932,  r = 0.813....significance...appropriateness of the level (comparing with the theoretical level), etc;

Response 8-9:

An r-value of 0.65 suggests a moderate positive correlation between variables. 

0.813 indicates a strong positive correlation between the  items,

an α-value above 0.70 is generally considered good. ,α=0.932 is relatively high and indicates a strong degree of internal consistency, which is typically desirable for a reliable measurement scale.

Comment 10: to make some corrections M 年龄 = 26.38?!

Response 10: I have changed this mistake.

Comment 11-12:

to mention the source of each Figure and each Table;

to explain the legend of Figure 2; even if it is mentioned in Figure 2- humanistic! There is not enough correlation with the explanations provided in the text;

Response 11-12:

I have added the source and unified the humanistic into cultural attractions.

Comment 13:to follow the journal's guidelines in citing the references in the text! The text's citing style does not respect the journal's citing guidelines.

Response13:

I have changed the references.

Reviewer 2 Report

Dear Authors,

Overall article focuses on the very important area and is well structured. There are some points that needs to be addressed to improve the article.

The article deals with the color’s hue, saturation and brightness properties. There is a need of colorful illustrations to describe different photos of tourist destinations with color attributes. May be its available in appendix A as mentioned on page 7 but not available for review.

The hypothesis statements may be revised to a clear singular statement and bifurcated such as H1a and H1b for natural destinations and cultural destination respectively. Similarly, H2 and H3.

The concept of borderline significance on page 9 and 10 needs to be clarified with references.

Regards

Author Response

Thank you very much for taking the time to review this manuscript. 

These are my responses to the comments:

Comment1:The article deals with the color’s hue, saturation and brightness properties. There is a need of colorful illustrations to describe different photos of tourist destinations with color attributes. May be its available in appendix A as mentioned on page 7 but not available for review.

Response1:I added an appendix at the end of the thesis.

Comment2:The hypothesis statements may be revised to a clear singular statement and bifurcated such as H1a and H1b for natural destinations and cultural destination respectively. Similarly, H2 and H3.

Response2: I have changed the hypothesis:

H1a: For nature-based tourist attractions, high saturation color photos can improve attractiveness ratings.

H1b: For cultural-based attractions, the effect between saturation color photos and attractiveness ratings disappears.

H2a: This effect is mediated by the psychological distance. For nature-based destinations, psychological distance perception reduces attractiveness.

H2b: For cultural -based destinations, the relationship between psychological distance and attractiveness is insignificant.

H3a: Geographic distance has a moderating effect. For non-local destinations, low saturation is more attractive for cultural attraction photos, and high saturation is more attractive for natural attraction photos.

H3b: For local destinations, the positive effect of high saturation on cultural and natural attraction photos disappears.

Comment3:The concept of borderline significance on page 9 and 10 needs to be clarified with references.

Response3: I have changed borderline significance to marginal significant. And I referenced "Pritschet, L., Powell, D., & Horne, Z. (2016). Marginally significant effects as evidence for hypotheses: Changing attitudes over four decades. Psychological science, 27(7), 1036-1042."

Round 2

Reviewer 1 Report

We appreciate that the authors addressed the recommendations but did not follow the journal's guidelines on how to mention the references in the text.

Sustainability | Instructions for Authors (mdpi.com)

"In the text, reference numbers should be placed in square brackets [ ], and placed before the punctuation; for example [1], [1–3] or [1,3]. For embedded citations in the text with pagination, use both parentheses and brackets to indicate the reference number and page numbers; for example [5] (p. 10). or [6] (pp. 101–105)."

Author Response

Thank you for your comments.

Comment1:We appreciate that the authors addressed the recommendations but did not follow the journal's guidelines on how to mention the references in the text.

Response1:We have changed the references In the thesis.